# Nucleic Acids Analytical Methods for Viral Infection Diagnosis: State-of-the-Art and Future Perspectives

**DOI:** 10.3390/biom11111585

**Published:** 2021-10-27

**Authors:** Emanuele Luigi Sciuto, Antonio Alessio Leonardi, Giovanna Calabrese, Giovanna De Luca, Maria Anna Coniglio, Alessia Irrera, Sabrina Conoci

**Affiliations:** 1Azienda Ospedaliero, Universitaria Policlinico “G. Rodolico-San Marco”, Via Santa Sofia 78, 95123 Catania, Italy; 2CNR-IPCF, Istituto per i Processi Chimico-Fisici, Viale F. Stagno D’Alcontres 37, 98158 Messina, Italy; antonio.leonardi@dfa.unict.it (A.A.L.); alessia.irrera@cnr.it (A.I.); 3Department of Chemical, Biological, Pharmaceutical and Environmental Sciences, University of Messina, Viale Ferdinando Stagno d’Alcontres 5, 98166 Messina, Italy; giovanna.calabrese@unime.it (G.C.); giovanna.deluca@unime.it (G.D.L.); 4Department of Medical and Surgical Sciences and Advanced Technologies “G.F. Ingrassia”, University of Catania, Via Sofia 87, 95123 Catania, Italy; ma.coniglio@unict.it; 5Istituto per la Microelettronica e Microsistemi, Consiglio Nazionale delle Ricerche (CNR-IMM), Ottava Strada n.5, 95121 Catania, Italy

**Keywords:** infections diagnosis, viral NA analysis, point-of-care, silicon-based technologies, microfluidics

## Abstract

The analysis of viral nucleic acids (NA), DNA or RNA, is a crucial issue in the diagnosis of infections and the treatment and prevention of related human diseases. Conventional nucleic acid tests (NATs) require multistep approaches starting from the purification of the pathogen genetic material in biological samples to the end of its detection, basically performed by the consolidated polymerase chain reaction (PCR), by the use of specialized instruments and dedicated laboratories. However, since the current NATs are too constraining and time and cost consuming, the research is evolving towards more integrated, decentralized, user-friendly, and low-cost methods. These will allow the implementation of massive diagnoses addressing the growing demand of fast and accurate viral analysis facing such global alerts as the pandemic of coronavirus disease of the recent period. Silicon-based technology and microfluidics, in this sense, brought an important step up, leading to the introduction of the genetic point-of-care (PoC) systems. This review goes through the evolution of the analytical methods for the viral NA diagnosis of infection diseases, highlighting both advantages and drawbacks of the innovative emerging technologies versus the conventional approaches.

## 1. Introduction 

The Coronavirus pandemic has recently focused the attention of researchers worldwide on infectious diseases, thus requiring the best screening strategy for safeguarding public health.

Infectious diseases are disorders provoked by a wide group of microorganisms including viruses, bacteria, fungi, etc., that under certain conditions become dangerous for human safety and capable of rapid transmission among humans or animals. This leads to a series of signs and symptoms that are mostly dependent on the type of microorganism causing the infection and the human district involved.

Viruses are submicroscopic microorganisms composed of two core elements: the nucleic acid genome, either double- (ds) or single-stranded (ss) DNA or RNA, and a protein-based shell called “capsid”. They exist in different habitats but are obliged intracellular parasites, thus they need to infect a living organism as a host for their growth and replication [1]. The infectious diseases caused by viruses usually respond to rest and home remedies but, for more severe cases, they need hospitalization and specific therapies. However, some types of infections have no valid treatments yet due to the microorganism drug resistance or its structural variability, as in the case of the human immunodeficiency virus syndrome, hemorrhagic fever caused by Ebola virus, and severe acute respiratory syndrome (SARS) caused by variants of the coronaviruses [2].

Therefore, in order to face the invasive transmission of some types of viruses, early detection and isolation of infected people is crucial to provide a strategy able to control the infection sources and appropriate treatment of the infecting agent.

In this review, a complete overview of the approaches developed for virus detection and infections diagnosis will be given, reporting the state-of-the-art of both the conventional methods and the advancements brought by the emerging new detection technologies.

## 2. Conventional Methods for Virus Detection

Virus detection for the diagnosis of infections is generally performed by an indirect or direct identification.

Methods based on indirect detection involve virus isolation through its introduction and proliferation into suitable host cells via traditional culture [3] or faster centrifuge-enhanced techniques [4,5]. Once proliferated, virus can be detected by the evaluation of morphological alterations and other cytopathic effects (CPEs) expressed by the infected cells or before cell damage by using intracellular staining of viral functional proteins (pre-CPE analysis).

Methods based on direct identification instead consist of straight virus detection from its biological source without pre-proliferation and propagation and are generally performed by immunological or molecular approaches.

The immunological identification uses antibodies as probes for the virus direct detection within a sample. Antibodies are available in polyclonal, monoclonal, and recombinant formats [6,7,8] and are used to interact specifically with a series of antigens exposed by the viral structural and functional proteins. The immune interactions can be achieved by different strategies, such as the blotting techniques for membrane-mediated interactions or the Enzyme-Linked Immunosorbent Assay (ELISA) for sandwich-type interactions [9]. These methods are generally not cost effective, and, above all, they must be carried out by specialized personnel in a qualified laboratory environment. Additionally, immunoassays do not achieve the same sensitivity as molecular detection methods [10].

Molecular methods for virus identification are based on the nucleic acid tests (NATs). These are multistep procedures where the pathogen NA is first extracted and purified from a biological sample (blood, urine, saliva, etc.), using different sample preparation kits, and then detected by PCR. This reaction allows amplification of a specific genetic sequence, unique to an individual organism, through the catalytic action of polymerase enzymes and thermal cycling. So far, the PCR reaction has been extensively studied and optimized, introducing the advancement of quantitative real time PCR (or qPCR), which allows quantitative evaluation of the amplified genetic target through fluorescent labels [11,12].

Although NAT has reached the greatest maturity in infectious disease diagnosis, conventional approaches are mostly still time-, cost- and labor-consuming, dependent on expensive machines (since the PCR reaction requires large and expensive thermal cyclers), and not suitable for miniaturized and decentralized analysis in the point-of-care format, since they are constrained to dedicated laboratories, thus becoming inadequate for the extensive use required during a pandemic outbreak of an infectious disease [13]. Figure 1 schematizes the main steps of the NAT analytical process.

Due to the above limitations, NATs are further evolving towards faster, portable, and integrated technologies called genetic point-of-care (PoC). 

## 3. NATs Improvements towards PoC Applications 

The term point-of-care (PoC) refers to all diagnostic tests that can be performed as close as possible to the patient, providing analytical results in a very short period for an immediate diagnosis and/or therapeutic decision [14]. The genetic PoC, in particular, must be able to manage, integrate, and merge the fundamental steps required by a complete molecular analysis (NA extraction, amplification, and detection) of a sample for a unique, portable, and user-friendly solution to be used outside a laboratory by unspecialized personnel. Therefore, in order to reach the right level of miniaturization, integration, and automation, evolution of conventional NATs for virus detection was mandatory.

Most of the improvements concerned the NA extraction step, focusing on the enhancement of the purification and isolation step, and the qPCR reaction, focusing on both the chemistry of probes used for the amplification/detection and the thermal cycling.

### 3.1. NATs Evolution: NA Extraction

NA extraction is a crucial step in molecular analysis as the efficiency of the entire NAT procedure depends on the quality of the genetic material isolated from the biological sample. Tissue biopsy, blood, urine, or saliva samples contain an amount of viral genome that is very low compared to the co-present human genetic material. Therefore, the purity of the extracted material must be very high, and an improvement of the purification and isolation steps is mandatory.

Over the last 20 years, solid phase extraction (SPE) has become the most common method for viral NA extraction due to some advantageous features, such as the minimal need for hazardous chemicals, reduced and easier manipulation, automation capability, and increased throughput [15]. SPE is based on the capture of a target genetic material from the lysed cell debris by binding/absorption on a solid surface (purification step) and the subsequent elution of purified NA using buffers with specific chemical properties (isolation step) [16].

SiO_2_ is one of the most used solid materials for SPE extraction since it is well known that, under proper ionic strength conditions, it specifically captures DNA. Actually, most of the current commercial extraction kits use silica in the form of (a) core-shell beads featuring a magnetic core covered by an SiO_2_ shell that are moved by external magnets, as in the case of the Magazorb kits [17] (Figure 2a); or (b) micro-filter mounted in a plastic column, such as the spin-column of the Qiagen kits [18] (Figure 2b).

Silicon is also a widely used material for SPE. It is very appealing for PoC system integration due to its optimal physical properties, including a low heat capacity, good thermal conductivity, and the possibility to incorporate electrode and microelectronics circuitry that imprint the so-called intelligence on board.

A drawback of the conventional SPE protocol, however, is that the chemical agents (i.e., alcohols and chaotropic salts) used during the extraction procedure can inhibit the subsequent qPCR reaction [19]. This implies several washing steps to remove all chemical traces and requires complex and expensive architectures for the fluid’s management.

Therefore, new technologies for SPE have been developed that focus on the silicon derivation, such as functionalization based on amino groups, chitosan, and graphene oxides [20,21,22,23,24], and patterned structures that increase the surface–area ratio (SVR), in order to enlarge the exchange areas used for the target NA binding/transport and allowing an excellent quality of separation and purification of the genetic material from the cell debris. Most of these advanced silicon-based technologies have been used in combination with microfluidics, which brings the advantages of volume downscaling, a reduction of the reagent and sample consumption, and a high degree of miniaturization and integration, suitable for PoC system development [25].

As an example, in 2017, Takano et al. reported an advanced and simplified SPE method based on low-cost polycationic silica particles for the enhanced extraction of free DNA from human urine [26]. The positively charged coated particles, prepared by mixing silica gel with polycationic polymer poly-Lys, were able to capture up to 1.3 μg of cell-free DNA with a negatively charged backbone from 50 mL of urine without the need for high concentrations of chaotropic agents, thus increasing the purity of the extracted DNA.

Birch et al. instead developed a dual porous silica (DPS) structure for SPE composed of two monolithic disks, synthesized from tetramethyl orthosilicate and potassium silicate, incorporated into a microfluidic device [27]. With this structure, the DPS was able to rapidly extract a large quantity of DNA from human urine samples (less than 35 min) with high integrity for downstream analysis, allowing SPE incorporation into the microfluidic device without leakage of the sample or reagents.

In 2017, Petralia et al. developed another example of a patterned silica structure in a microfluidic device for miniaturized SPE purification and isolation of viral NA [28]. Using the hepatitis B virus (HBV) genome as the analytical sample, a downsized (few centimeters) biofilter made of silicon micropillars, opportunely designed to increase the SVR and the amount of bound target NA, was fabricated and characterized. The pillars–NA interaction was mediated by a charge-based reversible adsorption occurring between the pillars’ superficial epoxydic groups and the negatively charged oxygens exposed by the NA backbone. The NA was first selectively linked to the pillars, washing away all the unbounded molecules including cells debris and other chemicals, and then eluted by deionized water. Thanks to the synergy of the absorption chemistry and the designed layout, the silicon pillars showed an extraction efficiency about 16% higher than that measured with commercial kits within a low-cost miniaturized device, providing an extraction strategy for genetic fully integrated PoC systems.

### 3.2. NATs Evolution: Redox PCR Probes

Once extracted and purified, the NA material is analyzed by molecular methods, such as qPCR.

A wide series of probes for qPCR are used to allow real-time detection of the genetic target during its amplification, such as oligonucleotide probes and intercalating agents. These agents, in particular, are able to enter the two paired bases of a double-stranded nucleic acid (dsNA) and interact with the minor groove [29]. Conventional and commercial intercalating probes, such as ethidium bromide, SYBR Gold, Evagreen, and SYBR Green, have a transduction active center that emits fluorescence once intercalated in the amplified product, producing the optical signal for the target sequence detection. This type of transduction, however, implies some limitations for integration into PoC systems since the conventional optical modules required for detection are not easily miniaturizable, are expensive and constrained to cooling supports, and the fluorescent probes are not stable enough to be used as reagents on-board [30].

To overcome the above reported limitation, redox-active compounds have been proposed as alternative intercalating probes for qPCR allowing electronic detection. Some of these are complex, composed of metals (such as osmium, ruthenium, etc.) that have been properly engineered to have the metallic center covalently bound to a planar intercalating ligand, such as phenanthroline, bypiridine derivates (BPY), or dipyridophenazine (DPPZ) [31,32,33,34]. As reported in Figure 3, the redox activity of these probes allows electrochemical detection in the NAT method that brings the advantages of a high level of miniaturizability and good sensitivity and robustness, which are ideal for integration into PoC systems.

In 2011, Limoges et al. used a series of metallic redox-intercalating compounds as probes for qPCR combined with electrochemical detection by Square Wave voltammetry and reported the osmium complex Os[(bpy)_2_DPPZ]^2+^ as the most performant, since it is chemically stable under qPCR thermal cycling, a strong intercalating agent towards the amplified dsNA, and able to enhance the sensitivity of the NAT method at about 10^3^ copies/reaction [35].

Another application of the redox probe for qPCR was presented in Petralia et al. in 2015 [36]. Again, they used the Os[(bpy)_2_DPPZ]^2+^ probe and developed a PCR-based electrochemical method for the detection of hepatitis-B virus (HBV) in a miniaturized silicon device, where 5’-thiolated oligonucleotides (used as primers) were chemically immobilized on the surface of a micro working electrode (WE) made of platinum with a density of about 4.0 × 10^12^ molecules per cm^2^ [37]. Thanks to the interaction between the viral DNA target and the anchored oligos and the subsequent intercalation of the osmium redox probe inside the amplified double-strand target, it was possible to detect the HBV DNA in solid state after 10, 20, 30, and 40 cycles of amplification with a huge increase of sensitivity with respect to the traditional qPCR systems. Moreover, this approach offered a high degree of versatility, since the technology can be employed to detect various double helix DNA targets by changing the specificity of the anchored oligos, and the possibility of miniaturization.

### 3.3. NATs Evolution: Isothermal PCR

Another improvement of the qPCR concerns the reduction of the thermal complexity by isothermal approaches.

Isothermal PCR is an alternative amplification method that uses a single temperature to rapidly and efficiently accumulate NA target sequences without the constraint of the thermal cycling [38,39]. One of the main features of this technology is the denaturation step of the dsNA target. Thanks to specific reagents (such as polymerase having strand-displacement ability, recombinase, and helicase), the double helix of the target is, in fact, enzymatically denatured, thus avoiding the use of high temperatures. Once denatured, the target is amplified by a combination of primers and extension phases that are set to be performed at the same temperature, thus excluding the need of a thermal cycling. This makes isothermal methods like Loop-Mediated Isothermal Amplification (LAMP), Recombinase Polymerase Amplification (RPA), and Helicase-Dependent Amplification (HDA) faster, as they do not need to reach a high number of thermal cycles but use continuous amplification, which provides detectable amplicons within a few minutes, which is more sensitive than qPCR, since they involve the synergy of multiple specific primers for the annealing step [40,41,42]. Schemes of the above-mentioned isothermal approaches are reported in Figure 4.

The isothermal amplification simplifies the architectural setup of the PCR-based method. In this case, in fact, since the thermal cycling is excluded, the temperature management does not need additional components (such as Peltier cells, resistors, capacitors, etc.) and high-power demand for the amplification process, thus reducing costs and architecture complexity. Moreover, such isothermal methods as LAMP and HAD can be coupled with an electrochemical detection that, as reported before, is perfectly suitable for integration into PoC systems [46].

## 4. Genetic PoC Systems for Viral Infection Diagnosis

### 4.1. PCR-Based Genetic PoC 

All the above reported NAT improvements were preliminary to the introduction of new functional modules for integration into genetic PoC systems, giving a full and accurate analysis of the viral NA in a sample-in-answer-out format to face the need for massive, fast, and decentralized diagnosis of infections. The development of integrated systems combining molecular detection with advanced extraction technologies for selective viral NA isolation has provided high-quality diagnosis of infectious disease. In fact, the possibility to specifically separate and extract few copies of viral genomes among more abundant copies of human or animal nucleic acid in a typical biological sample (blood, urine, saliva) has increased the amount of template available for the subsequent amplification and detection steps, providing an enhancement of the analytical resolution.

Examples of emerging PoC systems are reported in Figure 5. The synergy between microfluidics and silicon technology played a key role in the development of most of these systems, bringing a series of advantages in terms of reagent consumption optimization by volume reduction, incorporation of microelectronic circuits, and robustness of the material used. This led to the development of a series of genetic PoC systems for complete analysis of viral pathogens in infections [47]. A first product was the commercial GeneXpert (Cepheid) system, endorsed by the World Health Organization (WHO) in 2010, for the early detection of HIV-associated tuberculosis, reported in Figure 5a. The microfluidic device integrated a mechanical SPE extraction step, based on lysis by sonication and surface affinity, and PCR amplification and detection of the purified material, completing the analysis within 2 h [48]. A GeneXpert system was also developed for the severe and highly contagious Ebola virus disease (EVD), providing detection in 94 min with an LoD of 0.13 plaque-forming unit (PFU)/mL of Ebola Zaire virus [49]. This product was followed by others PoCs systems, such as the FilmArray (BioMerieux), combining NA extraction, based on SPE using silica-coated magnetic beads, and detection, by multiplex PCR, in a plastic cartridge of some tens of cm [50]. However, although performing all the main steps for a complete NA analysis, most of these PoC systems were just relocations of conventional NAT steps and required expensive and bulky instruments, thus leaving the integration and automation issue still unsolved.

Over the last years, many design and fabrication efforts have focused on improving the microfluidics of genetic PoC towards more integrated, automated, sensitive, and rapid devices. As an example, Sun et al. developed a genetic PoC system based on a portable silicon microfluidic chip for the live detection of virus in nasal swab samples [55]. Inside the chip, 10 microchannels with a deposited set of primers allowed an LAMP amplification of specific sequences of some equine viruses, such as equine herpesvirus 1 (EHV1) and equine influenza virus (EIV), and then optically detected by fluorescence imaging operated by a customized smartphone, revealing down to 5.5 × 10^4^ copies/mL of viral DNA. These performances make the system an example of genetic PoC for multiplex low-cost and portable analysis enabling remote diagnosis of viral coinfections for efficient epidemic surveillance. Rodriguez-Mateos et al. presented a fully integrated microfluidic PoC system for genomic SARS-CoV-2 RNAs analysis (Figure 5a), combining an NA extraction step based on immiscible filtration and surface tension (IFAST) with a detection via colorimetric reverse-transcription loop-mediated isothermal amplification (RT-LAMP) [51]. Within a downsized architecture, using a simplified detection of color change by the naked eye, the platform was capable of detecting up to 470 copies/mL of virus in saliva samples in about 1 h, with a 100% specificity, leading to a potential increase of the COVID-19 screening speed and an early detection prior to the viral transmission.

The shape of the analytical system represented an important issue in the microfluidics evolution, inspiring a lot of commercial products, such as the LIAT analyzer (IQuum), proposing a lab-in-a-tube technology for HIV detection [56], or the GenePoC system, reporting a downsized microfluidic cartridge for Influenza A and B virus detection [57]. A very appealing solution was introduced by the lab-on-disk technology. The disk shape allowed an improvement of PoC automation thanks to the possibility of rotation by centrifugal forces inducing fluid movements and other operations, such as liquid mixing, aliquot, switching, valving, and storage. In 2019, Zhou et al. developed a microfluidic-RT-LAMP chip system for the simultaneous detection of three types of coronaviruses (porcine epidemic diarrhea, porcine deltacoronavirus, and swine acute diarrhea syndrome-coronavirus) [58]. The system consisted in a polymethyl methacrylate disk containing a series of chambers (devolved to the sample storage, reaction, waste collection, etc.) for viral NA amplification by RT-LAMP and a detector integrating temperature control, high-speed centrifugation, and fluorescence reading and analysis, provided by iGeneTech (Ningbo, China). With this composition, the system was able to detect and quantify the NA of all three viral targets within 1.5 h, with a limit of detection (LoD) of about 1 × 10^2^ copies/mL, showing potential for rapid, sensitive, and high-throughput PoC diagnosis. In 2020, Sciuto et al. developed and characterized a lab-on-disk PoC technology for fully integrated viral NA analysis (Figure 5b) [52]. It was a hybrid structure composed of a plastic microfluidic disk containing all chambers required for the viral NA extraction and purification through SPE using magnetic beads (lysis, binding, washing, and elution), and a silicon module for NA amplification and detection by qPCR. The movements of beads, sample, and reagents were induced by a disk reader equipped with magnets and actuating all the centrifugal forces, thermal control, and optical fluorescence imaging required for NA analysis. Using the hepatitis B virus (HBV) DNA genome as the viral target, it was shown that the system was able to purify the viral NA from the starting sample and detect down to eight copies/reaction of the target, bringing an improvement in terms of automation and sensitivity with respect to the conventional qPCR platforms.

Microfluidics evolution paved the basis for the emerging droplet manipulation technology [59,60]. This strategy simplified the integration of microfluidic components in PoC systems by using droplets with embedded superparamagnetic particles and reagents/samples. By merging and splitting with each other through magnetic actuators, the droplets are able to achieve all the fluidic functions and operations required by the NA analysis, substituting the conventional virtual pumps, valves, mixer, SPE substrates, and PCR reactors. This was the case of PoC systems, such as the droplet-based platform developed by Pipper et al., which allowed the detection of the avian influenza virus H5N1 from a throat swab within 30 min [61]. 

Recently, the literature reported some evidence of PoC systems based on droplet manipulation combined with digital nucleic acid analysis, such as digital PCR (dPCR) and digital isothermal amplification. This method provides sensitive detection and precise quantification of target nucleic acids by the partitioning of the sample into droplets, each containing either zero or one (or, at the most, a few) template molecules. The single partition goes through an individual PCR reaction and, thanks to the fluorescence detection, is considered positive (1, fluorescent) if it contains target and negative (0) if not. The combination of droplets that tested negative or positive with the Poisson statistics allows estimation of the exact number of the NA target copies in each partition and calculation of the absolute template concentration in the original sample [62].The utility of the digital NA analysis has been demonstrated in many diagnostic applications. As an example, in 2021, Yu et al. proposed a self-partitioning SlipChip (sp-SlipChip) microfluidic device for the slip-induced generation of droplets to detect human papilloma virus (HPV) DNA by digital LAMP [63]. The chip was composed of top and bottom plates that can slip past each other, generating aqueous droplets used in the final fluorescent digital amplification, being capable of quantifying the viral DNA strain within a concentration range of 7.0 × 10^2^–1.4 × 10^7^ copies/mL. Recently, Xu et al. proposed an advanced spSlipChip microfluidic device combined with a portable integrated (PI) dPCR system, reported in Figure 5c, for quantitative analysis of the BK virus (BKV) genome in urine samples [53]. Through to a simplified slip-induced self-partitioning mechanism, enhanced by a series of microchannels and bridges, the SlipChip device was able to operate the formation of droplets, containing urine lysate sample and PCR mix, without a precise alignment of the contacting plates. A dedicated reader for both thermal control and fluorescence imaging, then, performed dPCR quantification of the BKV DNA with a dynamic range of 3.0 × 10^4^ to 1.5 × 10^8^ copies/mL within 2 h. 

The microfluidics evolution in fully integrated genetic PoC proceeded in parallel with the improvement of the structuring material used. The literature reports some PoC systems substituting silicon with other materials that have received particular attention due to their simplified, low-cost, and robust properties [64,65,66]. In 2020, Seok et al. proposed a paper-based PoC system reporting a fully integrated platform for the molecular diagnostics of three kinds of tropical viruses (Zika, dengue, and Chikungunya virus) in human serum, described in Figure 5d [54]. The system consisted of a chip made of paper containing a series of pads and dried chemicals for both the automatic fluidic flow-based extraction and the LAMP detection of viral RNAs. With this structure, the paper-based system was able to detect the target viruses in the range of 5–5000 copies/mL simultaneously and with extreme accuracy. Some evidence has also reported graphene as an alternative material for viral analysis, due to its unique optical and electrical properties, as from Liu et al., who developed a graphene conductive film performing highly sensitive detection of rotavirus [67], or Chen et al., who fabricated a graphene field-effect transistor (GFET)-based portable system for the detection of human influenza virus H1N1 [68,69].

### 4.2. PCR-Free Genetic PoC 

So far, virus detection in genetic PoC systems has mostly been based on NA amplification by PCR technologies. Despite its maturity, this approach implies several challenges, such as the selection of reagents with the right stability for an on-board usage (up to 1 year), the simplification of sample preparation and PCR chemistry to reduce the number of steps involved, and the integration of several modules for the thermal and opto-electrical control requiring certain energy consumption. Moreover, PCR may fail to amplify the target NA due to the high genetic variability of some viruses [70]. Since NA amplification was quite constraining, alternative PCR-free virus detection strategies, reported in Figure 6, have been proposed in genetic PoC.

Zhao et al., for example, developed a PCR-free system for the rapid detection of avian influenza virus H5N1 (Figure 6a) [71]. The system was a modified microarray platform with immobilized oligos used to directly and specifically hybridize the H5N1 RNA and, then, allow another poly-A-tailed intermediate oligos to form a sandwich complex with the target. This complex was bound by stained gold nanoparticles, giving a light scattering signal that was detected and quantified by a dedicated optical reader. The assay identified the H5N1 viral RNA with extreme accuracy, discriminating from other influenza virus subtyping (H1N1, H3N2), and velocity, since no NA amplification was required.

In 2017, Sciuto et al. proposed a technology for portable and rapid electrochemical detection of HBV without NA amplification (Figure 6b) [72]. The device consisted of a silicon miniaturized chip containing three microelectrodes and the chemical strategy employed was based on the hybridization between the target viral genome and two specific oligonucleotide probes grafted on top of the WE of the chip. Once formed, the hybrid complex was exposed to the redox-active compound Os(bpy)_2_DPPZ (see Section 3.2), releasing a current signal processed by a portable electronic board for Square-Wave voltammetry. With this structure, the system was able to detect and quantify the HBV genome without any preliminary amplification, reaching a low LoD of 20 copies of DNA analyzed. 

Similarly, Liu et al. developed a fully integrated and user-friendly PoC chip for the PCR-free detection of hepatitis C virus (HCV) in blood (Figure 6c) [73]. The chip used a passive fluidic method capable of first extracting the target RNA from blood and then detecting it by label-free charge-based electrochemical assay. Using specific PNA sequences anchored on top of nanostructured microelectrodes and the combined action of two redox-active probes (Ru(NH_3_)_6_^3+^ and Fe(CN)_6_^3−^), the chip was able to electrochemically detect the target RNA within 30 min without the need for amplification.

## 5. Conclusions

The growing demand for massive viral infections diagnosis, together with the low resource settings of certain infrastructures, has resulted in the need to develop increasingly more integrated, low-cost, and portable platforms for rapid and decentralized analysis. This also suggests the need to achieve early detection of viruses, improving the level of diagnostics. 

Virus detection has evolved over time, moving from conventional indirect identification by cultural evaluations of viral cytological effects in host cells to direct immunological detection of viral antigens and direct molecular analysis of the viral genome by NAT. This molecular approach provided the most accurate and complete analysis by viral NA extraction and its PCR detection. However, although it has reached the greatest maturity and the highest level of affordability, NAT is still constrained to dedicated laboratories and long and complex procedures so that considerable efforts, bringing new SPE and isothermal PCR methods for NA extraction and detection, have focused on performing NATs in more integrated and automated systems called genetic PoC.

In the last decade, PoC approaches have allowed complete viral molecular analysis in the sample-in-answer-out format to be achieved, reducing the cost, time, and complexity of conventional methods. The literature reported a series of genetic PoCs that, thanks to the synergy between the silicon technology and microfluidics and the incorporation of reagents and electronic circuits on board, that have evolved towards fully integrated, automated, and portable systems, as reported by the droplet manipulations-based technologies and lab-on-disk platforms. Moreover, new PCR-free PoC strategies have been proposed in order to improve NA detection in terms of time and cost consumption by excluding the amplification step.

Altogether, this evidence proves that the molecular analysis achieves the best level of accuracy and portability, providing new measures to face infection outbreaks and global pandemics.

## Figures and Tables

**Figure 1 biomolecules-11-01585-f001:**
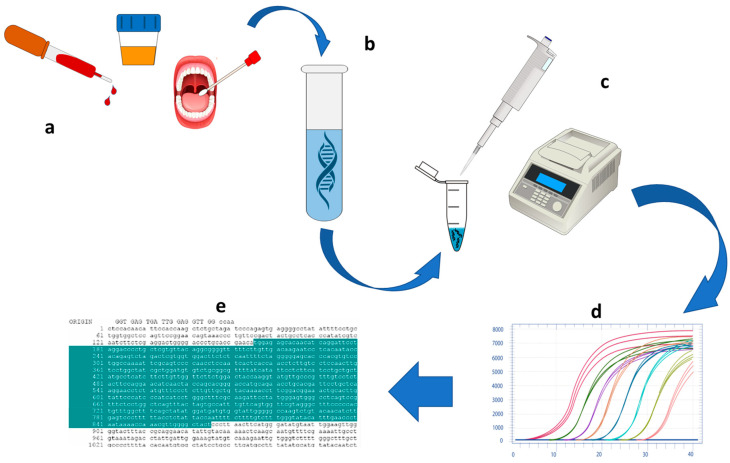
NAT analytical steps based on PCR: (**a**) sample collection; (**b**) NA extraction; (**c**) PCR amplification; (**d**) real-time PCR quantification; (**e**) sequence detection results.

**Figure 2 biomolecules-11-01585-f002:**
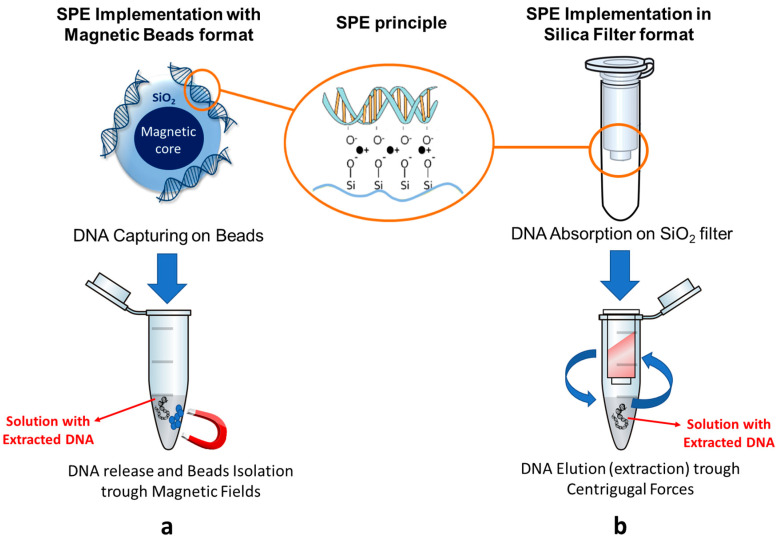
DNA absorption and isolation by solid phase extraction based on SiO_2_-coated magnetic beads and separation device (**a**) and with silica filters and spin columns through centrifugation (**b**).

**Figure 3 biomolecules-11-01585-f003:**
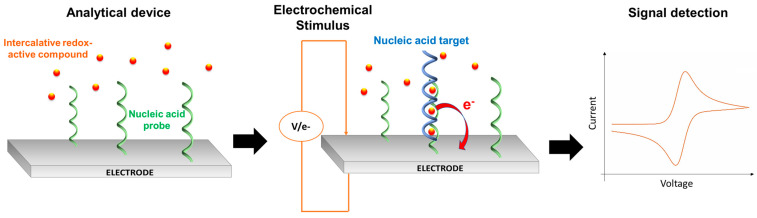
Generic scheme of nucleic acid analysis through intercalative redox-active compounds: detail of the electrochemical stimulus generation, by target–probe NA interaction and redox compound intercalation inside the dsNA, and the electrochemical signal detection.

**Figure 4 biomolecules-11-01585-f004:**
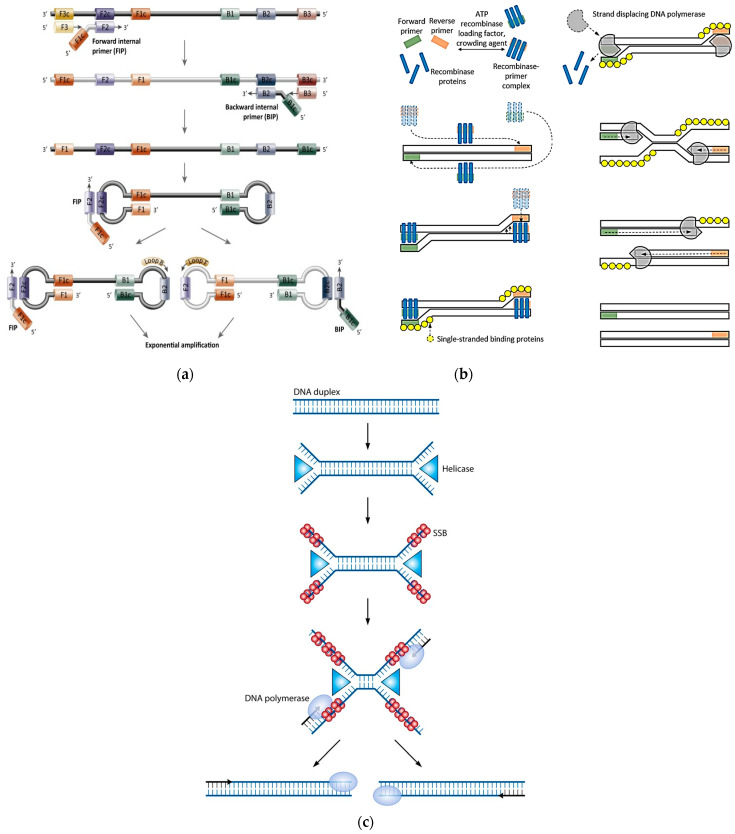
Isothermal approaches for NATs. (**a**) Loop-Mediated Isothermal Amplification. Reproduced with permission from [43]. Copyright 2021, Elsevier. (**b**) Recombinase Polymerase Amplification. Adapted with permission from [44]. Copyright 2021, Elsevier. (**c**) Helicase-Dependent Amplification. Reproduced with permission from [45]. Copyright 2021, American Society for Microbiology—Journals.

**Figure 5 biomolecules-11-01585-f005:**
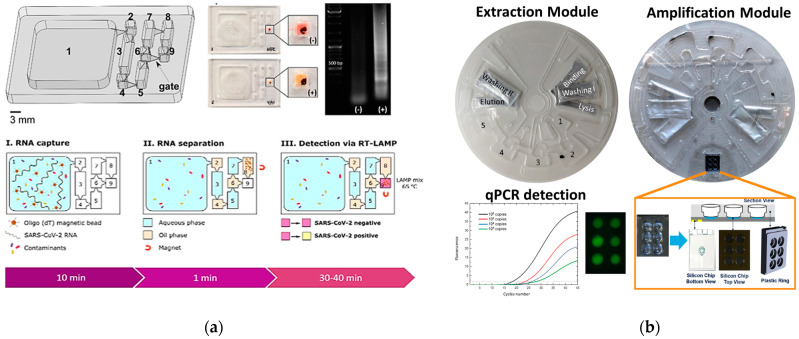
Schemes of the genetic PoC systems. (**a**) Fully integrated microfluidic PoC system for SARS-CoV-2 RNA analysis. Adapted with permission from [51] Copyright 2021, Elsevier. (**b**) Lab-on-disk for viral NA analysis. Adapted with permission from [52]. Copyright 2021, John Wiley & Sons—Books. (**c**) Sp-SlipChip microfluidic device and PI dPCR system for BKV DNA analysis in urine samples: detail of sample processing, droplet formation, and viral DNA detection by dPCR. Adapted with permission from [53]. Copyright 2021, Elsevier. (**d**) Paper-based integrated PoC for tropical virus diagnostics. Adapted with permission from [54]. Copyright 2021, Elsevier.

**Figure 6 biomolecules-11-01585-f006:**
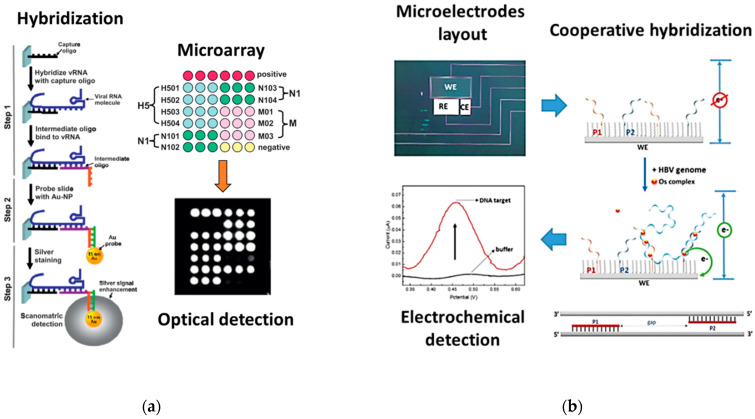
PCR-free virus detection strategies in genetic PoC systems. (**a**) Gold nanoparticle-based genomic microarray for the specific identification of avian influenza virus. Reproduced with permission from [71]. Creative Commons CC BY, Springer Nature. (**b**) Miniaturized electrochemical device for the PCR-free detection of HBV. Adapted from [72] (**c**) Fully integrated PoC system for PCR-free detection of HCV in blood. Adapted from [73] with permission from the Royal Society of Chemistry.

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
