# Peer review of "Nucleic Acids Analytical Methods for Viral Infection Diagnosis: State-of-the-Art and Future Perspectives"

_biomolecules, 2021, doi:10.3390/biom11111585_

Round 1

Reviewer 1 Report

The manuscript entitled ‘Nucleic acids analytical methods for viral infections diagnosis: 2 state-of-the-art and future perspectives.’ summarizes basic methods used for analysis nucleic acid, including those appropriate for virus identification.

The topic is important and, because of the high attendance for viruses, could be very useful. However, I feel imbalance and/or incompleteness among the methodological descriptions.

In section 2, blotting and ELISA are mentioned as cheap methods, accessible to not specialized staff. On the other hand, PCR was written as not suitable method for miniaturized and decentralized analysis. I think it is not true and these features depend on the type of the method. There are a number of simple and fast solutions for PCR, but depending on the antibody need and sample number, ELISA and blotting may be really expensive. Immunochromatographic methods may be referred as simple solutions . Dedicated stuff is always needed as well, qualified to at least basic level, because of the often experienced contaminations.

In sections 3.1 and 3.2 there are more details about some techniques (redox probes), while others are only shortly mentioned (e.g. magnetic extraction). Furthermore, there are very simple and less useful figures (Fig. 1 and 2) that could be deleted or improved and complemented. The title of Fig. 1 is NAT analytical steps, but it is only one course, the PCR. In Fig. 1e it should be label that it is Sanger sequencing result, etc. The Fig. 2 and 3 is very simple and incomprehensible in itself.

Section 3.3: What are the detection possibilities after isothermal amplification?

Fig. 4: It would be important to improve the readability of the text.

At some sections the authors use ‘we’ instead of usage of passive sentence form. It would be more favourable the passive form.

No words about reverse transcription, blot types for nucleic acids, sequencing, gel electrophoresis (novel, quick, capillary methods included), digital PCR.

Section 4.1. One sentence might be integrated: why is important to develop virus specific systems, including nucleic acid isolation methods?

  • Page 1, line 43: viruses, bacteria…
  • Page 1, line 44: …human or animals.
  • Page 2, line 56: coronaviruses instead of Coronavirus

After refinement and completion, the review could be a good summary of this topic.

Author Response

Referee 1

The manuscript entitled ‘Nucleic acids analytical methods for viral infections diagnosis: 2 state-of-the-art and future perspectives.’ summarizes basic methods used for analysis nucleic acid, including those appropriate for virus identification.

The topic is important and, because of the high attendance for viruses, could be very useful. However, I feel imbalance and/or incompleteness among the methodological descriptions.

We thank the referee for this comment and revision. Answers point-by-point are reported below.

  • In section 2, blotting and ELISA are mentioned as cheap methods accessible to not specialized staff. On the other hand, PCR was written as not suitable method for miniaturized and decentralized analysis. I think it is not true and these features depend on the type of the method.

There are a number of simple and fast solutions for PCR, but depending on the antibody need and sample number, ELISA and blotting may be really expensive. Immunochromatographic methods may be referred as simple solutions. Dedicated stuff is always needed as well, qualified to at least basic level, because of the often experienced contaminations.

We thank the referee for this comment. We agree the ELISA methods can be also expensive. We, therefore, modified the text (lines 83-85) adjusting this point and highlighting that in any case these are methods that can be carried out in specialized lab environment.

About the PCR, we also agree with the referee that there are a number of simple and fast solutions for PCR, however at the moment there is no solution available in PoC format to be used by not specialized personnel. To better explain this point, we modified the text at line 98.

  • In sections 3.1 and 3.2 there are more details about some techniques (redox probes), while others are only shortly mentioned (e.g. magnetic extraction). Furthermore, there are very simple and less useful figures (Fig. 1 and 2) that could be deleted or improved and complemented. The title of Fig. 1 is NAT analytical steps, but it is only one course, the PCR. In Fig. 1e it should be label that it is Sanger sequencing result, etc. The Fig. 2 and 3 is very simple and incomprehensible in itself.

We thank the referee for the comment. According to your suggestion, we better described the magnetic extraction that is an option of SPE in section 3.1 (lines 135-140).

Figure 1 was improved substituting the (e) part with results coming from Real Time PCR method, reported in (d) part, instead of sequencing that actually created confusion in the scheme concept.

Figure 2 and 3 were also edited adding more details and complemented with additional descriptions.

  • Section 3.3: What are the detection possibilities after isothermal amplification?

We thank the referee for the comment. The isothermal amplification simplifies the setup of PCR-based method for the typical associated detection that is optical. In this case, in fact, since the thermal cycling is excluded, the temperature management doesn’t need additional components (such as Peltier cells, resistors, capacitors, etc.) and high-power demand for the amplification process, thus reducing costs and architecture complexity of the thermal management and detection optical system. Moreover, there are evidences in literature reporting some isothermal approaches (such as LAMP and HAD) combined to the electrochemical detection that, as is known, is perfectly suitable for portable analytical systems. The text at lines 252-257 has been modified including the above-reported considerations. In any case, the new perspectives for the PCR methods, after the new isothermal amplification, are those named PCR free that are described in section 4.2.

  • 4: It would be important to improve the readability of the text.

The Figure 4 has been enlarged in order to improve the text readability.

  • At some sections the authors use ‘we’ instead of usage of passive sentence form. It would be more favourable the passive form.

The passive form has been applied throughout the manuscript.

  • No words about reverse transcription, blot types for nucleic acids, sequencing, gel electrophoresis (novel, quick, capillary methods included), digital PCR.

We thank the referee for the comment. Even if reverse transcription, blot types for nucleic acids, sequencing, gel electrophoresis (novel, quick, capillary methods included) are well consolidated methodologies for NATs, however since we focused the topic of the manuscript on the evolution of NAT analysis towards PoC format, we preferred to treat specifically the PCR methods being those eligible for the evolution in PoC.

According to your comment, we improved the description of digital PCR by adding more details (lines 359-366).

  • Section 4.1. One sentence might be integrated: why is important to develop virus specific systems, including nucleic acid isolation methods?

We thank the referee for the comment. We improved the text on this point by adding the following paragraph in section 4.1 (lines 273-279): the development of integrated systems combining the molecular detection to advanced extraction technologies for the selective viral NA isolation is important towards the quality improvement of infectious diagnosis. In fact, the possibility to specifically separate and extract few copies of viral genomes among more abundant copies of human or animal DNA in a typical biological sample (blood, urine, saliva) increase the amount of template available for the subsequent amplification and detection steps, providing an enhancement of the analytical resolution.

  • Page 1, line 43: viruses, bacteria…
  • Page 1, line 44: …human or animals.
  • Page 2, line 56: coronaviruses instead of Coronavirus

All lines at pages 1 and 2 have been revised according to the referee.

Reviewer 2 Report

This is a comprehensive review on the different assay methods for detecting viruses in biological samples. The article is mostly focused on molecular techniques since those assays offer greater sensitivity despite being more expensive.  I do have a few comments.

  1. Although the figures are most appropriate, the text on many of them is so small that it is difficult to read.
  2. It would really be helpful to readers if there was a paragraph or 2 on the most common problems that need to be considered when selecting an assay.
  3. It was a bit surprising that there was no mention of the POC assays that were developed during the past Ebola outbreak, given that the whole process had to start from scratch
  4. There should be some brief discussion of looking for intact viral nucleic acids in stool samples or waste water effluent, as is being done for COVID.  Do any of the assays offer an advantage for this type of specimen?

Author Response

Referee 2

This is a comprehensive review on the different assay methods for detecting viruses in biological samples. The article is mostly focused on molecular techniques since those assays offer greater sensitivity despite being more expensive.  I do have a few comments.

We thank the referee for this comment and revision. Answers point-by-point are reported below.

  • Although the figures are most appropriate, the text on many of them is so small that it is difficult to read.

We thank the referee for the comment. We improved the figures of the manuscript by adding more details and enlarging the text.

  • It would really be helpful to readers if there was a paragraph or 2 on the most common problems that need to be considered when selecting an assay.

We thank the referee for the comment. Certainly, when selecting an assay there are some problems that are crucial for the final diagnostic results. However, since the topic of the manuscript was centred on the evolution of NAT technological aspects towards PoC format, we excluded most of the aspects related to the specific assay, focusing on the PCR methods being those eligible for the evolution in PoC.

  • It was a bit surprising that there was no mention of the POC assays that were developed during the past Ebola outbreak, given that the whole process had to start from scratch

We thank the referee for the comment. We improved the manuscript by adding the example of Cepheid PoC test for Ebola (lines 293-295).

  • There should be some brief discussion of looking for intact viral nucleic acids in stool samples or waste water effluent, as is being done for COVID. Do any of the assays offer an advantage for this type of specimen?

We thank the referee for the comment. Even if stool samples or waste water effluent are challenging specimens for NAT analysis representing important applicative aspects, however, as above mentioned, the manuscript focuses on the main technological advancement related to NATs methodologies describing the basic methods for both extraction and detection that can be of course implemented for several applications. In this view both stool or waste water samples are certainly to be carefully considered when this new NATs technologies will be implemented in the specific applications.

Round 2

Reviewer 1 Report

Some more comments to the manuscript (Please, check the text for other grammatical mistakes and refine the professional terms, if needed):

  • Page 3, line 3: Please, insert here the abbreviation (PoC)
  • Figure 1 legend: real-time PCR
  • Figure 2 legend:
    • solid phase extraction with lowercase
    • The sentences are too complicated. I would change it (just a recommendation), e.g. DNA absorption and isolation by solid phase extraction (a) based on SiO2 coated magnetic beads and separation device (b) and with silica filters and spin columns through centrifugation.
  • Page 7, line 303-308: Please, refine the sentences for better understanding. I feel some things are blurred.
    • ’ …simplifies the setup of PCR-based for the typical associated detection that is optical.’ – Which technology do you mean associated with optical detection for?
    • ’…thus reducing costs and architecture complexity of thermal management and detection optical system.’ – What do you mean here about the optical system? For which technology?
  • Page 8, line 331: diagnosis of infections
  • Page 8, line 333-334: high quality diagnosis of infectious diseases.
  • Page 8, line 335: nucleic acid, or DNA and RNA instead of DNA
  • Page 9, line 350: hours instead of h
  • Page 10, line 438: …of droplets tested negative or positive…

Author Response

Some more comments to the manuscript (Please, check the text for other grammatical mistakes and refine the professional terms, if needed):

We thank the referee for the comments. Point-by-point answers are reported below.

  • Page 3, line 3: Please, insert here the abbreviation (PoC)

We thank the referee for this comment. The abbreviation has been inserted as suggested.

  • Figure 1 legend: real-time PCR

We thank the referee for this comment. Figure 1 legend has been revised accordingly.

  • Figure 2 legend:
    • solid phase extraction with lowercase
    • The sentences are too complicated. I would change it (just a recommendation), e.g. DNA absorption and isolation by solid phase extraction (a) based on SiO2 coated magnetic beads and separation device (b) and with silica filters and spin columns through centrifugation.

We thank the referee for these comments. Figure 2 legend has been revised following the referee suggestions.

  • Page 7, line 303-308: Please, refine the sentences for better understanding. I feel some things are blurred.
    • ’ …simplifies the setup of PCR-based for the typical associated detection that is optical.’ – Which technology do you mean associated with optical detection for?

We thank the referee for the comment. We better clarify this point in the text as follows:

The isothermal amplification simplifies the architectural setup of PCR-based method.

    • ’…thus reducing costs and architecture complexity of thermal management and detection optical system.’ – What do you mean here about the optical system? For which technology?

We thank the referee for the comment. We better clarify this point in the text as follows:

In this case, in fact, since the thermal cycling is excluded, the temperature management doesn’t need additional components (such as Peltier cells, resistors, capacitors, etc.) and high-power demand for the amplification process, thus reducing costs and architecture complexity.

  • Page 8, line 331: diagnosis of infections

We thank the referee for the comment. May be there is a mistake with the lines numbers reported. The sentence was modified at line 274.

  • Page 8, line 333-334: high quality diagnosis of infectious diseases.

We thank the referee for the comment. The sentence was modified at line 276.

  • Page 8, line 335: nucleic acid, or DNA and RNA instead of DNA

We thank the referee for the comment. May be there is a mistake with the page and lines numbers reported. The sentence was modified at line 278.

  • Page 9, line 350: hours instead of h

We thank the referee for the comment. The term has been revised elsewhere.

  • Page 10, line 438: …of droplets tested negative or positive…

We thank the referee for the comment. The sentence has been revised at line 362.